# Genetic Disorders with Predisposition to Paediatric Haematopoietic Malignancies—A Review

**DOI:** 10.3390/cancers14153569

**Published:** 2022-07-22

**Authors:** Aleksandra Filipiuk, Agata Kozakiewicz, Kamil Kośmider, Monika Lejman, Joanna Zawitkowska

**Affiliations:** 1Student Scientific Society, Laboratory of Genetic Diagnostics, Medical University of Lublin, A. Racławickie 1, 20-059 Lublin, Poland; aleksandrafili6@gmail.com (A.F.); agatha096@gmail.com (A.K.); kamilkosmider96@gmail.com (K.K.); 2Laboratory of Genetic Diagnostics, Medical University of Lublin, A. Racławickie 1, 20-059 Lublin, Poland; monikalejman@umlub.pl; 3Department of Pediatric Hematology, Oncology and Transplantology, Medical University of Lublin, A. Racławickie 1, 20-059 Lublin, Poland

**Keywords:** genetic predisposition, cancer susceptibility, germline mutations, paediatric cancer, genetic counselling

## Abstract

**Simple Summary:**

A significant progress in understanding the genetic basis of childhood cancers has been made over the past years. Genome sequencing has identified significant differences between paediatric and adult cancers. A higher prevalence of germline alterations in cancer predisposition genes is observed, in comparison to somatic mutations in paediatric cancers. Interestingly, pathogenic germline variants and somatic mutations can affect the same genes. The early recognition of genetic predispositions for childhood cancers may provide an opportunity of therapy adjustment and specific screening for different syndrome-related malignancies. Research on the biological function of gene variants with increased cancer predispositions is critical to the needs of adequate patient management. Genetic counselling with targeted therapy have become basis of integrated cancer care in paediatric oncology.

**Abstract:**

The view of paediatric cancer as a genetic disease arises as genetic research develops. Germline mutations in cancer predisposition genes have been identified in about 10% of children. Paediatric cancers are characterized by heterogeneity in the types of genetic alterations that drive tumourigenesis. Interactions between germline and somatic mutations are a key determinant of cancer development. In 40% of patients, the family history does not predict the presence of inherited cancer predisposition syndromes and many cases go undetected. Paediatricians should be aware of specific symptoms, which highlight the need of evaluation for cancer syndromes. The quickest possible identification of such syndromes is of key importance, due to the possibility of early detection of neoplasms, followed by presymptomatic genetic testing of relatives, implementation of appropriate clinical procedures (e.g., avoiding radiotherapy), prophylactic surgical resection of organs at risk, or searching for donors of hematopoietic stem cells. Targetable driver mutations and corresponding signalling pathways provide a novel precision medicine strategy.Therefore, there is a need for multi-disciplinary cooperation between a paediatrician, an oncologist, a geneticist, and a psychologist during the surveillance of families with an increased cancer risk. This review aimed to emphasize the role of cancer-predisposition gene diagnostics in the genetic surveillance and medical care in paediatric oncology.

## 1. Introduction

Neoplasms develop as the after effect of an increase of acquired and physical genetic variations in proto-oncogenes, tumour-suppressor genes, and DNA-repair genes (deoxyribonucleic acid) [1].

Alfred Knudson, the father of modern cancer genetics and heritable predisposition, hypothesized that “retinoblastoma is a form of cancer caused by two mutational events. In the dominantly inherited form, one mutation is inherited via the germinal cells and the second occurs in somatic cells. In the nonhereditary form, both mutations occur in somatic cells” [2]. The past decade, with large-scale genomic analyses, gave a possibility for better understanding of genetic disorders with predisposition to paediatric cancer. However, the prevalence of cancer predisposition mutations among children is still incompletely known and is reported in 7–10% of paediatric cancer patients [3,4,5]. Germline mutations are genetic variants that occur in gametes and are present in every cell of an offspring. Cancer predisposition syndrome (CPS) is inherited when a germline mutation is present in the cancer predisposition gene. Somatic mutations can occur in all other cell types and their accumulation can lead to tumourigenesis. Paediatric cancers typically have a higher prevalence of germline mutations in cancer predisposition genes when compared with somatic mutations. Including the unique combination of somatic alterations on the background of inherited and de novo germline variants during diagnosis is crucial for the understanding of the development of paediatric cancer [6]. Zhang et al. analysed over 500 genes in relation to cancer predispositions. Mutations in 60 genes have been associated with clinically relevant autosomal dominant CPSs [7]. Notably, there are some CPSs with the same mutated genes as those targeted by somatic mutations, e.g., *PAX5* in acute lymphoblastic leukaemia (ALL) blasts [8]. Including somatic genetic information in clinical care has allowed for the development of precision medicine by targeted agents, such as tyrosine kinase inhibitors for children with ALL whose blasts harbour mutations of the Janus Kinase/Signal Transducer and Activator of Transcription (JAK-STAT) pathway. Furthermore, germline genetic information from children with tumours should be obtained to identify the group with an increased risk of therapy-associated toxicities, second malignancies and non-oncologic manifestations [6].

The very wide range of potential cancer-related genes and gene variants of unknown significance (VUS), which can have unexplored neoplastic potential, underlines diagnostic difficulties. Therefore, there is a need for multidisciplinary cooperation between a paediatrician, an oncologist, a geneticist, and a psychologist during the surveillance of families with increased cancer risk. This review aims to emphasized the role of cancer-predisposition gene diagnostics in the genetic surveillance and medical care in paediatric oncology.

## 2. Paediatric Cancer Incidence

Cancer is the leading cause of death in both adults and children; however, when absolute values are looked at, they are relatively low in children [9]. Since the 1980s, a continued upward trend in the age-standardized incidence rate of registered cancers in children aged 0–14 years from 124.0 to 140.6 per million person-years has been observed using the World Standard Population (WSP). Sub-Saharan Africa is an exception, where this trend has been decreasing. According to completed data for 2001–2010 from the International Agency for Research on Cancer, leukaemias are the most common cancers in children aged 0–14 years, followed by central nervous system (CNS) tumours and lymphomas. Leukaemia is diagnosed in approximately 31% of children with neoplasms [10]. In children aged 15–19 years, lymphomas are most frequently diagnosed, followed by epithelial tumours and melanoma. The highest morbidity was observed in children aged 0–4 years and 15–19 years. [10]

Children with cancer who live in high-income countries have good outcomes, with approximately 80% surviving 5 years after their diagnosis. However, more than 90% of children at risk of developing childhood cancer each year live in low-income and middle-income countries, where effective care is not broadly accessible [11,12]. Children from low-income and middle-income countries (LIMCs) have been noted to have the highest aged-standardized mortality rate, i.e., 5.9–7.4, compared with high-income countries (2.7) [11]. The association between cancer incidence and socio-economic status is not so unambiguous, especially in East Asia and Eastern Europe. It can partially be explained by the accumulation of pathological germline mutations in cancer predisposition genes. Other factors contributing to this trend are the differences in access to health care, diagnostic capacity, and environmental exposures.

## 3. Syndromes Predisposing to Haematological Malignancies

The estimated risk of the development of haematological malignancies among different genetic disorders is significantly variable. For instance, patients with Nijmegen breakage syndrome (NBS) have a cumulative cancer incidence amounting to over 70% by the age of 20 years [13,14].

Germline genomic investigations have significantly increased the knowledge of germline predisposition disorders that increase the risk of haematopoietic malignancies. Due to advances in the recognition of germline genetic variants in patients with malignancies, the World Health Organization (WHO) has classified myeloid neoplasms with germline predisposition as a new entity in 2016 [15,16]. The diagnosis of a familial myeloid malignancy has become crucial in providing care for this specific patient group. Myeloid neoplasm classification includes the following sections: (i) myeloid neoplasms with germline predisposition without a pre-existing disorder or organ dysfunction, (ii) myeloid neoplasms with germ line predisposition and pre-existing platelet disorders, and (iii) myeloid neoplasms with germline predisposition and other organ dysfunctions. For the purpose of this manuscript, the above major categories have been reviewed in relation to novel findings concerning germline predispositions in haematological malignancies. In addition, hereditary predispositions also include lymphoid and plasma-cell cancers, with recent discoveries of pathogenic variants in the *KDM1A/LSD1* and *DIS3* genes, respectively [17].

In this section, hereditary syndromes with organ dysfunctions predisposing to haematological malignancies have been reviewed. The role of DNA instability syndromes, inherited bone marrow failure syndromes (IBMFS) and immunodeficiencies has been highlighted. DNA instability syndromes are heterogenous disorders caused by constitutional pathogenic variants in genes encoding key proteins involved in DNA replication and the cellular response to DNA damage. The potential vulnerability of affected patients to oncogenesis is caused by (i) DNA damage, especially during attempted B- and T-cell receptor rearrangement, immunoglobulin class switching and somatic hypermutation; (ii) reduced immune repertoire leading to both infections and tumourigenesis, and (iii) impaired immune development with potential premalignant clonal selection [18]. This group includes ataxia telangiectasia, Nijmegen breakage syndrome, Bloom’s syndrome, xeroderma pigmentosum, constitutional mismatch repair deficiency, followed by ligase 1 and 4 deficiencies. The next group is individuals with IBMFS, such as Fanconi anaemia (FA), and telomere syndromes, such as dyskeratosis congenita and BMF associated with an *MYSM 1* mutation. Primary immune deficiency diseases (PIDs) cannot also be denied in relation to the development of haematological malignancies. In this review, PIDs with syndromic features are mainly reported.

### 3.1. DNA Repair Disorders

#### 3.1.1. Ataxia Teleangiectasia (A-T)

A-T is caused by bi-allelic pathogenic variants in the *ATM* gene located on chromosome 11q22.3-23.1. *ATM* encodes a cell cycle checkpoint, ATM kinase, belonging to the phosphoinsitidyl 3-kinase-related protein kinase family, which plays a role in the cell cycle and regulates the functions of multiple proteins, including suppressor proteins p53 and BRCA1, and the checkpoint kinase CHEK2 [19]. ATM is also involved in oxidative stress and mitochondrial metabolism [20].

Patients with A-T present with oculocutaneous telangiectasia, progressive cerebellar ataxia, choreoathetosis and immunodeficiency. Neurological features are usually observed in early childhood when children begin to sit or walk. However, children do not develop in the same manner and the diagnosis of A-T can be delayed to early school years when neurological features and telangiectasia worsen [21]. There are two forms of A-T categorized as classic (typical, early onset) and a mild form classified as atypical (variant, late onset) [21]. Clinically significant immunodeficiency may be associated with a risk of cancers, particularly lymphoid malignancies. Patients with immunodeficiency with a hyper IgM phenotype with hypogammaglobulinemia and patients with IgG_2_ deficiency showed decreased survival rates when compared to those with normal IgG [22]. Similarly, Suarez et al. reported an increased cancer risk in patients with a profound IgA deficiency [20].

The estimated cancer risk is 40% for the most common B-cell type non-Hodgkin lymphoma (NHL), Hodgkin lymphoma (HL), and T-cell ALL [20]. Lymphomas and leukaemias usually occur in individuals with classic A-T under the age of 20. Moreover, solid tumours, such as ovarian and breast cancers, gastric cancers, melanomas, and sarcomas are also described [19]. Marabelli et al. estimated the risk of the development of breast cancer in ATM mutation carriers at 6% by the age of 50 [23].

#### 3.1.2. Nijmegen Breakage Syndrome (NBS)

NBS develops due to a mutation affecting the *NBN* gene located on chromosome 8q21. The *NBN* gene encodes for nibrin, a component of the MRN complex (MRE11-RAD50-NBN), which is crucial in maintaining chromosomal integrity through DNA double-strand break repair, DNA recombination and cell cycle checkpoint control. Moreover, nibrin is involved in lymphocyte maturation by V(D)J recombination and immunoglobulin class switching [19]. Most patients are diagnosed with five base pair deletions in the *NBN* gene (c.657_661del5), which is a founder mutation in Slavic populations [24]. NBS is observed worldwide; nevertheless most patients come from Poland and Russia [13,14].

The characteristic clinical features of NBS include progressive microcephaly, dysmorphic facial features (including sloping forehead, prominent nose, long mandible, long philtrum), mild growth delay and premature ovarian insufficiency. Despite the presence of microcephaly at birth or shortly thereafter in 80% of patients with NBS, the average delay of syndrome diagnosis was 6.5 years [25]. A profound immunodeficiency of both cellular and humoral responses is observed in most patients. In addition to mutations in the *NBN* gene, an increased susceptibility to lymphoid malignancies can be explained by affected T-cell development with the presence of senescence signs in circulating T-lymphocytes expressing CD57, KLRG1, and PD1 [26]. Moreover, impaired telomeric repair with telomere attrition and a statistically lower total antioxidant status influences the carcinogenesis [27]. NHLs and T-ALLs are most frequently diagnosed among lymphomas and leukaemias, respectively. NBS patients are also sometimes diagnosed with medulloblastomas, rhabdomyosarcomas, gliomas, papillary thyroid carcinomas, gonadoblastomas, meningiomas, neuroblastomas and Ewing sarcomas [24]. In recent cohort studies, the role of haematopoietic stem cell transplantation (HSCT) has been evaluated. NBS patients with diagnosed cancer who received HSCT had a significantly higher 20-year overall survival (OS) than those who did not (42.7 vs. 30.3%, respectively). However, pre-emptive transplantation in NBS patients did not have a significant influence on 20-year OS [14].

#### 3.1.3. Bloom’s Syndrome (BS)

BS results from homozygous pathogenic variants mostly include missense mutations in the *BLM* gene located on chromosome 15q26.1, which encodes DNA helicase, called RecQ, which attaches and unwinds the DNA double helix and maintains genomic stability during the DNA copying process by limiting sister chromatid exchanges (SCEs) [28]. The clinical diagnosis can be confirmed by cytogenic analysis identifying an increased number of SCEs [29]. Less than 300 cases of BS have been reported worldwide and one third of the reported patients were Ashkenazi Jewish, due to the founder allele [30]. From the age of two, BS patients present with a significant sensitivity to sunlight characterized by erythematous rash in butterfly distribution. The typical phenotype also includes pre- and postnatal growth deficiency, a short stature, high-pitched voice and distinctive facial features with a narrow face, a small lower jaw, and prominent nose and ears. Most children with BS are vulnerable to common infections of the ear, nose and throat (ENT) areas and the gastrointestinal tract. Moreover, decreased fertility in males and insulin resistance are also observed [30]. Cancers are the most frequent complications in patients with BS. Although the wide distribution of neoplasms resembles that in the general population, they occur at much earlier ages and simultaneously. By the year 2018, there were 226 malignant neoplasms identified in 145 persons in the Bloom’s Syndrome Registry. Leukaemia and lymphoma were mostly reported [31]. Affected children are also diagnosed with gastrointestinal, genital, and urinary tract carcinomas, sarcomas, Wilms tumours, medulloblastomas and retinoblastomas [31]. Moreover, BS patients are prone to develop second malignancies, with colorectal cancer being the most common [19].

#### 3.1.4. Constitutional Mismatch Repair Deficiency (CMMRD)

CMMRD, a highly penetrant syndrome inherited in an autosomal-recessive manner, develops due to biallelic germline mutations in the MMR genes (*PMS2, MSH6, MSH2,* and *MLH1*), which control DNA replication fidelity with exonuclease domains in the DNA polymerases [32]. The hallmarks of replication repair deficiency are point mutations and microsatellite instability (MSI). CMMRD is mainly caused by mutations in *PMS2* occurring in almost 60% of reported families [33]. Interestingly, heterozygous monoallelic germline loss-of-function mutations in the MMR genes are observed in Lynch syndrome, called hereditary non-polyposis colorectal carcinoma (HNPCC).

Despite the diagnostic challenge due to clinical overlap between CMMRD and neurofibromatosis type 1 (NF1), a few signs are strongly associated with CMMRD. Patients with CMMRD are observed for café-au-lait spots, skin hypopigmentation, mild defects of immunoglobulin class switching, agenesis of the corpus callosum, or pilomatricomas (pilomatrixomas, calcifying epitheliomas of Malherbe) [34]. Café-au-lait spots observed in CMMRD patients have a slightly diffuse appearance, in contrast to those observed in NF1, which are more sharply delineated [35]. Adenomatous polyps in the small and large intestines are frequently observed during childhood [36]. There is an increased risk of multiple cancers that occur synchronously or metachronously. About one third of the CMMRD patients develop leukaemias or lymphomas as a primary or secondary malignancy. The most often reported malignancies are T-cell NHL, followed by T-cell ALL and AML [35]. CMMRD patients are also diagnosed with brain tumours and gastrointestinal cancers. Most brain tumours are high-grade gliomas with characteristic giant cells on histology [37].

#### 3.1.5. Xeroderma Pigmentosum (XP)

XP is caused by autosomal recessive inheritance of pathogenic variants in nucleotide excision repair (NER) genes, which can be assigned to seven complementation groups —XP-A to XP-G—and one variant form (XPV) [38]. XP-A is the most affected complementation group, followed by XP-C, representing 30 and 27% of all XP patients, respectively [39]. The NER is essential for the repair of ultraviolet-induced DNA damage. A founder mutation in XPA can be found in the Japanese population, followed by XPC in the Northern African population [40].

An increased sunlight sensitivity is observed in only 60% of XP patients. However, all XP patients suffer from early hyperpigmentation and an early onset of premature skin aging. The most serious problem due to the defective NER is UV-induced DNA damage accumulations which result in photo-carcinogenesis. The age of diagnosis of skin cancers in XP patients is statistically lower, in comparison to the general population, 8 and 60 years, respectively [41]. Other cancers described in XP are leukaemia, brain and spinal cord tumours, and other solid tumours [28].

### 3.2. Bone Marrow Failure

#### 3.2.1. Fanconi Anaemia (FA)

FA is a phenotypically and genetically variable disorder which is characterized by the different expressivity of multiple congenital anomalies and a risk of bone marrow failure [42]. It results from mutations in up to 22 *FANC* genes (most commonly *FANCA*, *FANCC* and *FANCG*) that coordinate DNA interstrand crosslinks (ICLs) [43].

The most common symptoms are haematological abnormalities, including cytopenias or BMF [43]. FA is clinically characterized by a short stature, abnormal thumbs, and café-au-lait spots. Severe phenotypes of FA can be observed during infancy as combinations of vertebral anomalies, anal atresia, cardiac malformations, tracheal-–esophageal fistula with oesophageal atresia, and structural renal and limb abnormalities (VACTER-L) [43]. However, up to 25% of patients can only be diagnosed with cytopenia, due to a normal phenotype [44]. BMF is observed during childhood in most patients. Squamous cell carcinomas of the head, neck, and anogenital regions are the most common solid tumours and AMLs the most common leukaemias. Interestingly, the highest risk of AML has been found in patients with no congenital anomalies [44].

#### 3.2.2. Dyskeratosis Congenita (DC)

DC, also known as Zinsser–Engman–Cole syndrome, is caused by mutations in several telomerase-shortening genes with multiple inheritance patterns, including X-linked *DKNC1* (Xq28), autosomal dominant *TERT* (5p15.33), *TNF2* (6p21.33), or autosomal recessive *NOP10* (15q14), *NHP2* (5q35.3), *TCAB1* (17p13.1) [28]. The evaluation of leukocytes by fluorescence in situ hybridization (FISH) showed telomere shortening. Due to the most commonly X-linked recessive inheritance, males are affected three times more frequently than females. Disease anticipation is associated with the progressive shortening of telomeres [45].

The cardinal dermatologic findings are reticulate hyperpigmentation, nail dystrophy and oral leucoplakia, commonly seen on the tongue [46]. Reticulate hyperpigmentation with atrophy, and poikiloderma with telangiectasia usually appear during late childhood. Skin lesions are commonly observed on sun-exposed areas [47]. Up to 90% of patients with DC can also be affected by BMF. The lungs may also develop fibrosis. BMF, followed by myelodysplastic syndrome (MDS), or leukaemia are the most frequent causes of death [46].

#### 3.2.3. Shwachman Diamond Syndrome (SDS)

SDS is mainly caused by mutations in the SBDS gene located on 7q11.21. Recently the DnaJ heat shock protein family member C21 (DNAJC21), elongation factor-like 1 (EFL1) and signal recognition particle 54 (SRP54) have been associated with an SDS-like phenotype. All the involved genes participate in ribosome biogenesis including ribosome maturation process [48].

SDS is characterized by multiple organ involvement, including haematological disorders, metaphyseal chondrodysplasia, pancreas insufficiency and developmental delay [48]. Skeletal dysplasia and generalized osteopenia are common. Persistent or intermittent neutropenia, which results in recurrent infections, is a common finding usually seen some time before SDS diagnosis [49]. SDS patients have a high risk of MDS followed by AML [50].

#### 3.2.4. Diamond Blackfan Anemia (DBA)

DBA is a ribosomal disorder mostly caused by pathogenic variants including RPS19, RPS24, RPS17, RPL5, RPL11, and RPL35A [51].

DBA is characterized by craniofacial anomalies, such as a cute snub nose and wide-spaced eyes, followed by hypoplastic thumbs. Cardiac anomalies, such as an atrial and/or ventricular septal defect and the coarctation of the aorta are also observed. In the majority of patients, DBA is diagnosed before the first birthday, with the median age of 2 months, based on signs of severe macrocytic or normocytic anaemia and reticulocytopenia [52]

DBA patients have an increased risk of developing haematological malignancies (MDS, AML), followed by colon carcinoma, osteosarcoma, and urogenital malignancies [53]. In the DBA registry, the cumulative incidence of solid tumours and AMLs was approximately 20% by the age of 40. Cancer risks appear lower in DBA than in FA or DS [54].

#### 3.2.5. GATA2 Deficiency

GATA2 haploinsufficiency is caused by missense mutations or deletions in the *GATA2* gene located on chromosome 3q21.3, which is a critical transcriptional regulator of haematopoiesis, interacting with *RUNX1*, among others, to regulate haematopoietic stem cell survival. Moreover, *GATA2* is a regulator of genes involved in lymphatic valve morphogenesis [55]. The initial haematological presentation can be very variable, ranging from non-syndromic to syndromic cases with mycobacterial infections, monocytopenia, B-lymphopenia and pulmonary alveolar proteinosis in MonoMAC syndrome or lymphoedema, cutaneous warts and sensorineural deafness in Emberger syndrome. These patients also have a marked phenotypic corelation with MDS and AML [56]. Progression to AML can be associated with monosomy 7. Human papillomavirus and Epstein-Barr virus infections cause additional neoplasms [57].

### 3.3. Immunodeficiencies with Associated or Syndrome Features

#### 3.3.1. Cartilage–Hair Hypoplasia (CHH)

CHH is caused by variants in the non-coding RNA gene *RMRP* localized on 9p13 (RNA component of the mitochondrial RNA-processing endoribonuclease) [58]. Most reported CHH patients come from Amish and Finnish populations [59].

The main clinical features include metaphyseal dysplasia, a disproportionate (short limb) short stature, sparse and thin hair, and Hirschsprung disease [58]. The study performed on the Finnish population revealed a seven-fold increase of cancer incidence when compared to the general Finnish population. Patients with CHH mainly have an increased risk of NHL and basal cell carcinomas. Moreover, lymphoproliferative disorders, such as HL or ALL, are also diagnosed. [60].

#### 3.3.2. Wiskott-Aldrich Syndrome (WAS)

WAS is an X-linked immunodeficiency syndrome which results from mutations in the *WAS* gene localized on chromosome Xp11, which encodes WASP protein, acting as a key regulator in the signalling and movement of actin filaments in the cytoskeleton. WASP also supervises the proper function of T-cells and natural killers (NK) [61].

Due to a wide range of gene mutations, the disease has variable clinical presentations, ranging from a severe phenotype (classic WAS) to milder ones (X-linked thrombocytopenia and X-linked neutropenia). WAS patients have characteristic clinical features, including petechiae, prolonged bleeding, haematemesis, melaena due to thrombocytopenia (present from birth), eczema and recurrent bacterial, viral, and fungal infections. Malignancies are most frequently observed in males at a median age of 9.5 years [62]. B-cell NHL, usually diffuse large B-cell lymphoma (DLBCL) (often Epstein-Barr virus positive), is the most common. Leukaemia, HL, other NHLs, such as Burkitt lymphoma, are also observed [63].

#### 3.3.3. SAMD9 and SAMD9L Syndromes

SAMD9 and SAMD9L syndromes are caused by heterozygous missense mutations with a gain-of-function in *SAMD9* and *SAMD9L,* which are located side by side on chromosome 7q21 and play a role in endosome fusions [57]. In recent studies, an aberrant karyotype with del (7q) has been widely described and can be considered pathognomonic for SAMD9/9L syndromes [64]. Moreover, the somatic-acquired diploidy of the distal 7q region by segmental uniparental disomy (UDP) has been reported due to proliferative advantage [65].

Germline variants in *SAMD9* are associated with a constellation of symptoms described by the acronym MIRAGE: Myelodysplasia, life-threatening and recurrent Infections, Restriction of growth, Adrenal hypoplasia with insufficiency, Genital abnormalities (46 XY females, bifid shawl scrotum, testicular dysgenesis, intra-inguinal or abdominal testes, clitoromegaly), and Enteropathy (enteropathy, reflux, achalasia) [64]. Additional features include prematurity, chronic lung disease, and developmental delay [66]. At the same time, germline variants in *SAMD9L* are linked to progressive neurological phenotypes and pancytopenia, Ataxia Pancytopenia (ATXPC) [64]. Moreover, a haematological phenotype is the most common in both syndromes, which plays a role in MDS and AML [64,65].

### 3.4. RASopathies

RASopathies are a group of syndromes characterized by the dysregulation of signalling through the RAS pathway, which is known to play a major role in lymphangiogenesis.

#### 3.4.1. Noonan Syndrome

NS is caused by mutations altering genes encoding proteins in the RAS–MAPK pathway, leading to dysregulation usually due to enhanced signal flow by this pathway [67]. Approximately 50% of NS patients have a pathogenic missense variant in *PTPN11* located on the chromosomal band 12q24.1, followed by germline mutations of *SOS1*, *RAF1*, *RIT1*, *BRAF*, *KRAS*, *MAP2K1*, *MRAS*, *NRAS*, *RASA2*, *RRAS2*, *SOS2*, or *LZTR1* [68].

NS is characterized by distinctive facial features (low-set, posteriorly rotated ears, down slanted palpebral fissures, hypertelorism, epicanthal folds and ptosis), a short stature, developmental delay, learning difficulties, congenital heart defects, most frequently pulmonary valve stenosis or hypertrophic cardiomyopathy, and renal anomalies. NS is one of the most common syndromic causes of congenital heart disease [69]. In addition, up to 80% of males with NS have uni- or bilateral cryptorchidism. Children with NS have an increased risk of cancer development, including juvenile myelomonocytic leukaemia (JMML), acute myelogenous leukaemia, or B-cell ALL. Cases of embryonal rhabdomyosarcoma, granular cell tumour, pilocytic astrocytoma, Sertoli tumour in cryptorchid testis, and glial tumours have also been reported [67].

#### 3.4.2. Neurofibromatosis Type 1 (NF1)

NF1 is an autosomal dominant genetic disease, which is caused by a mutation of *NF1* localized on chromosome 17q11.2, which encodes neurofibromin. Neurofibromin is a cytoplasmatic tumour-suppressive protein that negatively regulates the RAS signalling. Complete penetrance is sometimes observed, although its expression is extremely variable. A significant genotype–phenotype correlation has been identified in NF1 p.Met992del, Nf1 p.Arg1809, NF1 microdeletions, and missense mutations in NF1 codons 844–848 [70]. The phenotype is triggered by miscellaneous factors, such as age-dependent manifestations, the timing and number of second hits in specific cells, allelic and nonallelic heterogeneity, epigenetics, modifying loci and environmental factors.

NF1 has a limited clinical presentation in the early years of life and there are many conditions which should be considered in differential diagnosis. Clinical manifestations depend on the patients’ age. Patients with NF1 typically have café-au-lait spots, Lisch nodules, and axillary or inguinal freckles. Moreover, motor/speech delays, autism spectrum disorders and scoliosis are also observed [71]. Pinti et al. estimated the efficiency of the National Institutes of Health clinical diagnostic criteria. Only 53% of children with clinically diagnosed NF1 had confirmed pathogenic NF1 variants. Legius syndrome, NF2, MEN2B and LEOPARD syndrome have clinically mimicked NF1. In contrast, 40% of cases with genetically confirmed NF1 had no clinical diagnosis [72]. Therefore, Legius et al. made an attempt to revise the NIH criteria of NF1. During diagnosis, at least one or two café-au-lait spots or freckling should be bilateral. Choroid abnormalities were included into criteria. Moreover, identification of NF1 variant alone in unaffected tissue is not the basis of diagnosis [73].

Patients have an estimated 60% risk of developing cancers, in comparison to the general population [71]. NF1 patients have a significantly elevated risk of JMML and account for 10% of children with this type of leukaemia [74]. NF 1 is also associated with a highly increased risk of malignant peripheral nerve sheath tumour (MPNST) and rhabdomyosarcoma, followed by optic pathway glioma (OPG) and pilocytic astrocytoma. Up to 50% of individuals diagnosed with MPNST have underlying NF1 [75].

#### 3.4.3. Casitas B-Lineage Lymphoma (CBL-Syndrome)

CBL syndrome is also known as Noonan syndrome-like disorder with or without juvenile myelomonocytic leukaemia [76]. CBL syndrome is caused by germline heterozygous mutations, usually of the missense type, in the *CBL* gene located on 11q23.3, which encodes the CBL protein. This E3 ubiquitin–protein ligase is involved in cell signalling and the ubiquitination of proteins [77].

CBL patients usually present phenotypic features of Noonan syndrome including short stature, facial dysmorphism, hyperpigmented nevi, pterygium colli, cardiovascular abnormalities, pectus excavatum, joint laxity, cubitus valgus, neonatal feeding problems, and attention deficit hyperactivity disorder [78]. CBL mutations have to be included in the differential diagnosis of foetal pleural effusions, hydrops fetalis, and foetal nuchal oedema [76]. There is a significantly increased risk of JMML development [79].

### 3.5. Aneuploidies

Constitutional aneuploidies including Down syndrome (DS), constitutional trisomy 8 mosaicism (CT8M) and Klinefelter syndrome (KS) are representatives of chromosomal anomalies associated with haematological malignancies. Children with constitutional trisomy 21 (DS) have a significantly higher risk of acute leukaemia; the incidences of AML and B-cell ALL in children with DS are 150 times and 33 times higher, respectively, than in the general population of the same age [80]. Myeloid leukaemias of DS are classified as ML-DS and, immunophenotypically, they are erythron–megakaryoblastic leukaemias presenting with thrombocytopenia and/or myelodysplasia [81]. Between 5 to 30% of myeloid leukaemias are preceded by the pre-leukaemic syndrome, transient abnormal myelopoiesis (TAM), which is caused by the concomitance of trisomy 21 and truncating variants in the *GATA1* gene [80]. CT8M is a phenotype characterized by clinical features including elongated facial features, abnormally shaped ears, strabismus, camptodactyly, clinodactyly, deep plantar and palmar skin furrows, vertebral/hip anomalies, and cardiovascular and urogenital malformations. CT8M has been reported as one of the most common abnormalities in malignant myeloid disorders, such as AML, MDS and myeloproliferative neoplasms (MPNs) [82]. KS is mainly caused by the karyotype with an additional X chromosome. There is no clearly defined association between cancers and 47, XXY patients, and the underlying mechanisms of an increased cancer risk still need to be investigated [83]. Studies among the Swedish cohort revealed about a three-fold increase in haematological malignancies when compared to the general population. Leukaemias and NHL were mainly diagnosed [84]. However, the prevalence of KS among males with B-ALL does not seem to be higher than in the general population [85].

## 4. Non-Syndromic Germline Variants Predisposing to Malignancies

A high index of clinical suspicion in the recognition of genetic predispositions to haematological malignancies is crucial to provide high quality patient care. Particular attention is especially needed when dealing with patients with no typical family history and no noticeable disease-associated signs and symptoms. In this section, non-syndromic germline predispositions to paediatric haematological malignancies were reviewed.

### 4.1. Germline Predisposition without a Pre-Existing Haematological Disorder or Organ Dysfunction

#### 4.1.1. Familial AML with CEBPA Mutation

*The CEBPA* gene is located on chromosome 19q. Mutations occur in two main hotspots: the N-terminal (frame-shift mutations) and C-terminal (frame insertions/deletions) [86]. *CEBPA* encodes the myeloid transcription factor CCAATT, expressed in myelomonocytic cells, which is an important mediator of granulocytic maturation, partnering other master regulators of haematopoiesis, such as RUNX1 or GATA factors [87].

Biallelic *CEBPA* mutations with N-terminal frameshift *CEBPA* germline mutation followed by acquired C-terminal somatic mutation as second event have been observed in more than 10% of patients diagnosed with AML [88]. AML is usually diagnosed at 25 years of age without previous cytopenias. Over 50% of patients with familial AML develop AM recurrence [87].

#### 4.1.2. Familial MDS/AML with Mutated DDX41

*DDX41* is located on chromosome 5q35 and encodes CEBPAα, an ATP-dependent nucleoid acid helicase [89].

Biallelic mutations in affected patients are associated with a significantly increased risk of MDS/AML. Moreover, lymphomas, CML and MM are also observed [90]. In addition, affected individuals were found to be haploinsufficient with a 5q deletion, involving the *DDX41* locus and associated with responses to lenalidomide [91].

#### 4.1.3. PAX-5-Associated Leukaemia Predisposition

Germline mutations in *PAX-5* located at 9p13, which encodes the DNA-binding transcription factor involved in maturation of B-cells, have been reported to increase the risk for precursor B-ALL [57]. Leukaemic cells have typically been diagnosed by the somatic loss of the wild-type PAX5 allele, either by formation of iso-/dicentric 9q chromosomes or deletions of 9p [56]. The risk of leukaemia development decreases after the first decade [57].

#### 4.1.4. IKZF1 Susceptibility to ALL

*IKZF1,* located on 7p12.2, encodes the founding member of the IKAROS family of zinc-finger transcription factors and plays a critical role in the regulation of lymphoid development [92].

Germline *IKZF1* alterations can be associated with common variable immunodeficiency and IgA immunodeficiency [56]. Germline *IKZF1* variants have also been reported in ALL. Moreover, germline variants influence the response of leukaemia cells to both conventional chemotherapeutic agents and kinase inhibitors [92].

#### 4.1.5. DICER1 Syndrome

The *DICER* gene located on chromosome 14q32.13 encodes a cytoplasmic endoribonuclease Dicer protein, which regulates the expression of cellular microRNA (miRNA) by splitting precursor molecules into miRNA. Mutations in the *DICER1* gene affect the processing of miRNA with the subsequent disruption of gene expression control, including a loss of function in tumour supressors or gain of function in oncogenes [93].

Loss of *DICER1* typically presents as pleuropulmonary blastoma, the most common lung tumour of infancy and early childhood. DICER1 syndrome seems to be associated with rare forms of T-cell phenotype classical HL [94]. CNS manifestations are pituitary blastoma, pineoblastoma, or DICER-associated spindle cell sarcoma [95]. Close follow-up should be considered with focusing on changes in respiratory function, menstruation, and a development of thyroid goitres due to potential malignancy [96].

Next-generation sequencing has also identified other pathogenic variants, such as *ERCC6L2* associated with AML, *SRP72* with MDS, and *MBD4* with early-onset AML [17].

#### 4.1.6. Li-Fraumeni Syndrome (LFS)

LFS is a cancer predisposition syndrome caused by pathogenic germline variants in the *TP53* tumour suppressor gene on chromosome 17p13 [97]. The p53 protein normally acts as a guardian of the genome. If DNA damage occurs, p53 triggers a response based on the transcription regulation of numerous genes involved in the cell cycle, DNA repair, apoptosis, senescence, and cellular metabolism [98]. The penetrance of pathogenic/likely pathogenic (P/LP) variants of *TP53* is variable. The p53 proteins bearing missense mutations classified as dominant–negative, which are highly penetrant, are usually detected in the families of those with childhood cancers, in contrast to null variants with a lower penetrance [98,99]. The difference is statistically significant with the mean age of tumour onset of 23.8 years in individuals harbouring missense mutations when comparing to 28.5 years in those with null mutations [99].

Cancers in individuals with LFS generally occur in age-related phases. LFS is particularly characterized by neoplasms classified as core tumours in Chompret Criteria—breast cancer, soft tissue sarcoma, osteosarcoma, brain tumour, and adrenocortical carcinoma [100]. Choroid plexus carcinoma and low hypodiploid B-ALL are highly correlated with LFS. MDS and AML are also observed. Germinal variants identified in children who do not fulfil the LFS genetic testing criteria have changed the diagnostic approach to the carriers of mutations in the *TP53* gene. The broad spectrum of cancers justifies the concept of LFS as a heritable *TP53*-related cancer syndrome (hTP53rc) [98,99,100]. Bougeard et al. updated the diagnostic Chrompret Criteria, which were included in a broader spectrum in Riperger et al., too [99,101].

### 4.2. Germ Line Predisposition with a Pre-Existing Haematological Disorder

#### 4.2.1. ETV6-Related Familial Neutropenia (Thrombocytopenia, Type 5)

The *ETV6* gene, which is located on chromosome 12p13.2, encodes a master hematopoietic transcription factor that is a part of a large family comprising 28 members involved in angiogenesis and haematopoiesis, as well as the growth and differentiation of cells. *ETV6* is one of the most rearranged genes in ALL and MDS, due to more than 30 reported translocation partners. In addition, somatic rearrangements to *RUNX1* are observed in one quarter of children with ALL [102] However, studies using cord blood from healthy newborn children indicate that *ETV6–RUNX1* translocations may occur at a rate of 1% or more in the healthy population [103].

Families with *ETV6* germline syndrome are typically diagnosed with thrombocytopenia, causing usually mild to moderate bleeding episodes. It is estimated that leukaemia develops in up to 30% of carriers, with ALL the most frequently observed, and MDS, AML, mixed phenotype acute leukaemia, DLBCL, myeloproliferative disease and plasma cell myeloma also observed [56].

#### 4.2.2. Familial Platelet Disorder with Predisposition for AML

The familial platelet disorder with a predisposition for AML is caused by inherited mutations in the hematopoietic transcription factor *RUNX1* located on chromosome band 21q22. *RUNX-1* mutant cells have defective hematopoietic differentiation, with reduced hematopoietic progenitors and abnormal differentiation of megakaryocytes [55]. The most common cause of tumorigenesis is the somatic acquisition of a second *RUNX1* mutation, followed by somatic mutations in the *CDC25C* gene and the acquisition of additional mutations in *GATA 2* [55].

Patients have a tendency to experience mild to moderate bleeding. The risk of malignant transformation into MDS and AML is estimated at approximately 35% (ranging from 20 to 60%) [42].

#### 4.2.3. ANKRD26-Related Thrombocytopenia

*ANKRD26*-related thrombocytopenia is caused by germline mutations in the 5′ regulatory region of *ANKRD26* located on 10p12.1, which encodes a protein that promotes MAPK signalling and megakaryocyte development [56]. Moreover, mutations of the *ANKRD26* gene result in the loss of RUNX1 and FLI1 binding, which are responsible for *ANKRD26* silencing during the late stages of megakaryopiesis and blood platelet development. That leads to the persistent expression of ANKRD26 and a profound defect in proplatelet formation [104].

Clinically, ANKRD26-related thrombocytopenia is characterized by moderate bleeding and an increased risk of developing AML and MDS when compared to the general population [55].

Interactions between germline and somatic mutations during the development of haematological malignancies were presented in Figure 1.

## 5. The Importance of Identifying Genetic Predispositions to Paediatric Cancers

### 5.1. Newborn Screening for CPSs

The diagnosis of genetic syndromes and the care of children before the development of cancer is a crucial issue. Newborn population-based genetic screening may reduce deaths due to paediatric cancers and is cost-effective when compared to the standard treatment of cancers [105]. Yeh et al. evaluated universal newborn screening with a targeted next-generation sequencing approach for pathogenic germline variants in *RET*, *RB1, TP53, DICER1, SUFU, PTCH1, SMARCB1, PHOX2B, ALK, WT1*, and *APC*. In the cohort of 3.7 million newborns, the model estimated that 1803 children would develop a CPS- related malignancy before the age of 20, 13.3% of whom would be identified at birth as at-risk due to pathogenic/likely-pathogenic (P/LP) variant detection. When implementing surveillance guidelines among P/LP heterozygotes, the cancer deaths before the age of 20 would be reduced by 53.5% [106]. Cascade testing, as an extension of newborn screening, of first-, second-, and third-degree relatives may be a favourable strategy to achieve population-level benefits in identification of cancer susceptibility mutations [107]. It should be noted that most cancer deaths (80–89%) were observed among siblings of healthy newborns [108].

### 5.2. Identification of Children with Cancers and Probable CPSs Who Would Benefit from Genetic Counselling

The key problems of genetic counselling include identifying appropriate patients for evaluation, helping families with genetic testing decisions and options, determining appropriate clinical management, and incorporating current guidelines concerning hereditary cancer syndromes in paediatric care [21]. Another major challenge is the need for cancer genetic counsellors who specialize in paediatrics. The genetic counselling profession has grown by over 100% in the last ten years and is expected to grow another 100% over the next ten years [109].

Jongmans et al. created an easy-to-use selection tool (Figure 2) to identify patients with already diagnosed cancer who may benefit from genetic counselling. The selection tool includes family history, specific malignancies, multiple primary cancers, specific features, and excessive toxicity of treatment [110].

However, this questionnaire has some limitations. The family history cannot be the solo indication used to guide the provision of genetic testing. Zhang et al. reported that only 40% of patients with germline mutations that could be evaluated had a family history of cancer [7]. This could be explained by DNA alterations associated with lower penetrance, which decreases the likelihood that a person will develop features of the associated syndrome. Moreover, it may result from de novo variants, parental germline mosaicism, recessive inheritance, or the masking of hereditary syndromes in small and young families [5,111,112]. Specific features which should be taken into consideration are congenital anomalies, facial dysmorphisms, intellectual disability, aberrant growth, skin anomalies, haematological disorders, and immune deficiency. Gargallo et al. evaluated Jongmans et al.’s tool in relation to pathogenic mutations and found a sensitivity of 94% and a specificity of 77% in their cohort [3]. Jongmans et al.’s criteria modification by Ripperger et al. (with an added category of genetic tumour analysis which may reveal a defect suggesting a germline predisposition and expanded list of cancer types associated with CPS, presented in Figure 1) was found to have 100% sensitivity in this cohort [3,101]. Schwermer et al. also reported a significant impact of the CPS questionnaire on diagnosis improvement. CPS was diagnosed in 9.4% of children using the questionnaire, in comparison to 3% during the control period (*p* = 0.032) [113].

Goudie et al. also took a step toward better identification of CPSs and created the McGill Interactive Pediatric OncoGenetic Guidelines (MIPOGG) in the form of a smartphone/tablet application which can help paediatricians make decisions about genetic referral of children with neoplasms [114]. MIPOGG exhibited a favourable accuracy profile for CPS screening and reduced the time to CPS diagnosis. A number of 410 of 412 patients with cancers and diagnosed CPSs were correctly evaluated with MIPOGG [115]. Moreover, MIPOGG was favourable for the identification of patients with a high predisposition to second malignancies, due to a possibility of decision making based on highly intensive surveillance [113]. Byrjalsen et al. also highlighted the need of identifying adult-onset CPSs, due to future the possibilities of surveillance. The performed study revealed adult-onset CPSs in 4.5% of the subjects [4].

### 5.3. The Role of Novel Genetic Sequencing (NGS)

Genetic testing with NGS technologies, such as targeted cancer gene NGS, whole exome sequencing (WES), or whole genome sequencing (WGS) has allowed for a wider understanding of mutations related to neoplasms. The initial step, i.e., a selection of a proper specific genetic test, should include the choice of the genes of interest, testing methodology and validation, variant interpretation, and turnaround time. For instance, the familial variant testing of a single gene is used for the diagnosis of CPSs with known familial variants. WES/WGS is applied in patients with unclear cancer phenotypes or multisystem phenotypes, or to identify novel genetic associations during scientific research [111]. Another point is that NGS raises complicated challenges regarding informed consent and the return of results. Nowadays, there is an ongoing process of standardizing variant interpretation to improve the quality and guarantee better comparability. Zhang et al. provide a step-by-step protocol for clinical interpretation that will help with better understanding of variant classification [116]. According to the American College of Medical Genetics and Genomics (ACMG), gene variants should be classified as benign, likely benign, variants of unknown significance (VUS), likely pathogenic (LP) or pathogenic (P) [117]. VUS identification does not allow for the identification of a genetic predisposition to cancers. Patients diagnosed with VUS should be periodically re-evaluated because the interpretation of VUS may change over time. Similarly, an arising challenge of WGS is to separate important driver mutations from numerous non-functional passenger mutations arising in the non-coding genome, which is more than 50 times larger than the coding genome. Moreover, obtaining genomic information for certain conditions could lead to anxiety or negatively impact health behaviour or even physiology. The information which does not have a clinical impact may not be beneficial for the patient [118]. In the performed survey, the parents of children with cancers did not consider WES an ethically disruptive technology during conducted interviews [119]. Another important issue of NGS is turnaround time to obtain the results [120]. The reports of 24 studies using NGS techniques in children, adolescents and young adults with cancers showed that median turnaround time from study enrolment to case presentation at a precision medicine tumour board was between 6–120 days with the median of 35 days [112,121]. The observed differences can be influenced by the waiting time for bioinformation analysis. The need of the analysis of sequencing results by multidisciplinary precision medicine tumour boards also delays turnaround time, due to the schedules of consultations, e.g., weekly [5]. The use of machine learning in NGS programs is considered a way to make NGS cheaper, faster, and better. Unfortunately, the quality of the training and dataset validation is still insufficient to eliminate bias in the data used to construct the algorithms and bias in the algorithms themselves [118]. Therefore, the lack of accessibility to infrastructure with the necessary computational resources and continuing relatively high costs are still not uncommon.

## 6. Incorporating Molecular Findings into Clinic

### 6.1. Multidisciplinary Cooperation

Along with the development of precision medicine, a multidisciplinary molecular tumour board (Figure 3) which provides individual patient reports is a significant step in the management of paediatric patients. The key role of the board is to assess the feasibility of pursuing actionable findings and the evaluation of a possible matched treatment and/or clinical trial [120,121,122,123].

### 6.2. The Impact of Diagnosis of Germline Variants Predisposing to Malignancies on Clinical Management

A high percentage of reportable germline alterations in paediatric patients diagnosed with neoplasms, estimated between 6 and 35%, with an average of 12%, underlines the importance of performing germline NGS [120,121,122,123]. Haematological malignancies and solid tumours have a similar number of germline variants. *TP53* mutations are the most often reported. Recognizing *TP53* as driver mutations in various neoplasms demonstrates a significant clinical feasibility of these findings, due to a possibility of providing genetic counselling concerning the future cancer risk.

Furthermore, the identification of a germline mutation can lead to a change in donor selection for children with haematological malignancies requiring a bone marrow transplant. Marks and al. reported a patient with AML and persistent thrombocytopenia who was referred for stem cell transplant. During the screening work-up, her donor sister was diagnosed with mild thrombocytopenia. Constitutional WES identified a germline mutation in *RUNX1* leading to a change in donor selection and referral of the affected family members to the clinical genetic service. Additionally, germline findings can contribute to difficult treatment decisions, such as palliative care instead of intensive chemotherapy with curative intent [123].

Importantly, the use of radiotherapy for the treatment of primary neoplasms in children with CPSs poses a very high risk of developing secondary malignancies. Additionally, there are some CPSs classified as DNA instability syndromes such as NBS or A-T, in which ionizing radiation must be omitted during diagnostic procedures due to an increased sensitivity to ionizing radiation. MRI or ultrasound investigation should be performed instead of CT [28].

Patients with a genetic predisposition to haematopoietic malignancies should be under constant oncological surveillance. Taking care of these patients should include awareness of signs of leukaemia and morphological tests. In a meta-analysis study, over 50% of children with leukaemia presented with hepatomegaly, splenomegaly, pallor, fever, and bruising. Abdominal symptoms which are not typically included in cancer guidelines were reported in one third of the patients (anorexia/weight loss) [124]. The most common features of leukaemia are also frequently reported in many common, self-limiting diseases of childhood and pose challenges for front-line clinicians to distinguish them, especially in children with immunodeficiency syndromes. Moreover, T-ALL, which often develops as progression of T-cell lymphoblastic lymphoma presents with different symptoms, such as a bulky mediastinum associated with superior vena cava syndrome [125].

According to the recent recommendations of AACR Childhood Cancer Predisposition Workshop, there is a higher benefit in screening children at a greater risk for MDS, or AML that occurs in the context of MDS (e.g., FA, SDS, severe congenital neutropenia, GATA2 deficiency, familial monosomy 7, CEPBA-associated predisposition to AML) than those at a greater risk for rapidly evolving haematological malignancies, such as ALL or AML [57]. At the baseline of haematological surveillance, a complete blood count (CBC) with manual differential and a bone marrow aspirate/biopsy should be performed in all patients. Then the AACR recommends an annual surveillance testing with CBC at a minimum. Nevertheless, the intervals for screening of haematological malignancies are not as clear.

According to the Toronto Protocol, children with LFS should have CBC done every 3–4 months from birth [126]. Hence, according to the AACR, blood examination can be omitted [57]. In children with CMMRD, CBC should be repeated with a frequency of every 6 months from the age of one year [33]. In patients with NS due to *PTPN11* or *KRAS* mutations associated with myeloproliferative disorder (MPD)/JMML, 3- to 6-monthly CBC with spleen size assessment should be considered starting at birth and continuing until the age of 5 [69]. Regardless of the underlying genetic condition, if changes to abnormal values in blood count develop, CBC should be repeated within 2–4 weeks. If CBC worsens over two or more measurements, a bone marrow biopsy/aspirate should be performed [57].

The necessity of routine bone marrow screening poses many questions. According to Porter et al., an annual clinical bone marrow evaluation is recommended only for children with CPSs with a greater risk for BMF and/or MDS/AML, including FA, familial AML with CEBPA mutations, SDS, severe congenital neutropenia, or GATA2 deficiency [57].

Follow-up after the end of therapy of haematological malignancies in children with CPSs should be adapted according to generally accepted treatment protocols, including CBC evaluations every 3 months for the first year of follow-up, every 6 months for the second year, and then annually. Nevertheless, children with CPSs should have follow-up extended over 5 years or even life-long, due to the increased risk for the second haematopoietic malignancy. Annual surveillance testing including CBC should be done [57].

In some syndromes, due to an increased risk of the development of various neoplasms, additional management of these patients is needed. The most important extra medical examinations are presented in Table 1. Imaging is considered crucial in the surveillance of many CPSs. Recently, nonionizing radiation imaging options, such as whole-body magnetic resonance imaging (WBMRI), have become more important [126]. Among individuals with LFS, WBMRI allowed for the early diagnosis and evaluation of survival rates. Villani et al. observed that the 5-year overall survival was 88.8% (95% CI 78.7–100) in the surveillance group, in comparison to 59.6% (47.2–75.2) in the non-surveillance group (*p* = 0.01) [127].

The cost-effectiveness of pre-symptomatic cancer surveillance has been evaluated. Tak et al. assessed cost-effectiveness for germline pathogenic variants in *TP53* [128].

### 6.3. The Role of Targeted Therapy in Paediatric Cancers

The concept of precision medicine with the consideration of individual variability during designing therapy has been developed for many years. Hence, post-genome-sequencing-era discoveries provide renewed opportunities for personalizing care of children with malignancies.

An analysis of over 3500 cases of children, adolescents, and young adults in 24 studies revealed a high rate of actionable variants, significant in molecular-driven precision medicine [120,121,122,123]. However, comparing the results is complicated due to a lack of standardization of potentially druggable events (PDEs), and the various techniques of molecular profiling. Paediatric MATCH defined an alteration as a PDE only for available treatment in phase II of a clinical study. Then, actionable alterations are classified within scales of target prioritization algorithms [41]. Most precision medicine programs include children with refractory or relapsed malignancies. In all primary malignancies, PDEs have been estimated between 12 and 100%, with an average of 50% [120,121,122,123]. The low incidence of actionable alterations in some studies results from examination tumour samples from newly diagnosed malignancies without considering a high risk or relapsed/refractory cancers in inclusion criteria [120,121,122,123]. Notably, PDEs differ in primary and relapsed tumours. This highlights the need of profiling the current neoplasm when considering targeted therapies.

On average, 21.5% of patients (rates ranging from 2 to 54%) could receive targeted therapy after decision of molecular tumour boards [120,121,122,123]. Despite the high detection rate of potentially actionable alterations, not all patients receive treatment with targeted agents. Among the reasons for nonadherence to tumour board recommendations are inability to access the desired targeted agents and a lack of available clinical trials. In addition, many patients with relapsed/refractory diseases are treated according to proven standard relapse therapies. Among antitumoral targeted therapies there are molecular and epigenetic targets and immunotherapy. Among actionable alterations, molecular targets with associated pathways including MAPK signalling and cell cycle control are the most affected [120,121,122,123].

## 7. Conclusions

Every childhood cancer harbours a unique correlation of inherited and de novo germline variants and somatic alterations. However, the prevalence of genetic disorders in children diagnosed with cancers with no common or non-syndromic phenotype is still underestimated due to low awareness of the pre-existing conditions. Molecular profiling with detection of potentially druggable events reveals opportunities of molecular-driven precision medicine. Improvement of diagnostic accuracy and outcomes, finding therapeutically actionable alterations are key steps for the management of children with cancers.

## Figures and Tables

**Figure 1 cancers-14-03569-f001:**
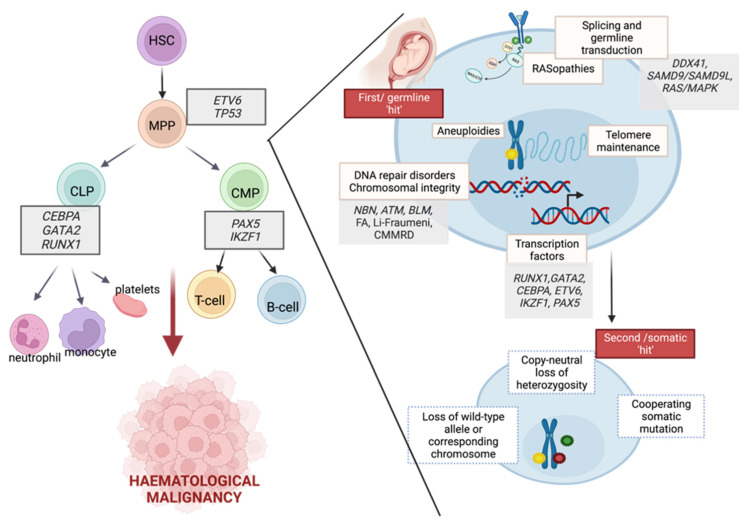
The role of germline predispositions in the development of haematological malignancies.. Hematopoietic development and associated germline alterations are presented to highlight the predisposition to neoplasms in myeloid and lymphoid lineages. Germline alterations in a multistep mutational process are presented as the first hit. Subsequently, many somatically acquired secondary events may promote a transformation that leads to the development of hematopoietic malignancies. HSC—haematopoietic stem cell, MPP—multipotent progenitor, CLP—common lymphoid progenitor, CMP—common myeloid progenitor. Image created with BioRender.com, accessed on 13 April 2022.

**Figure 2 cancers-14-03569-f002:**
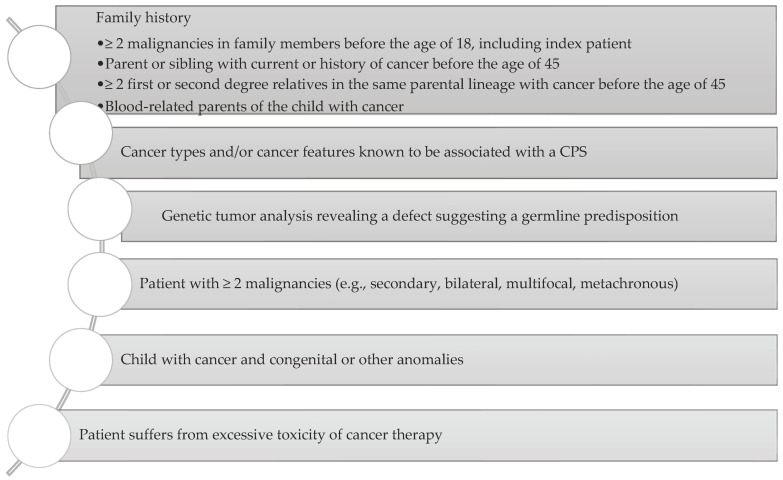
Consideration of a possibility of CPS diagnosis. Adapted from Ripperger et al., 2016 [101].

**Figure 3 cancers-14-03569-f003:**
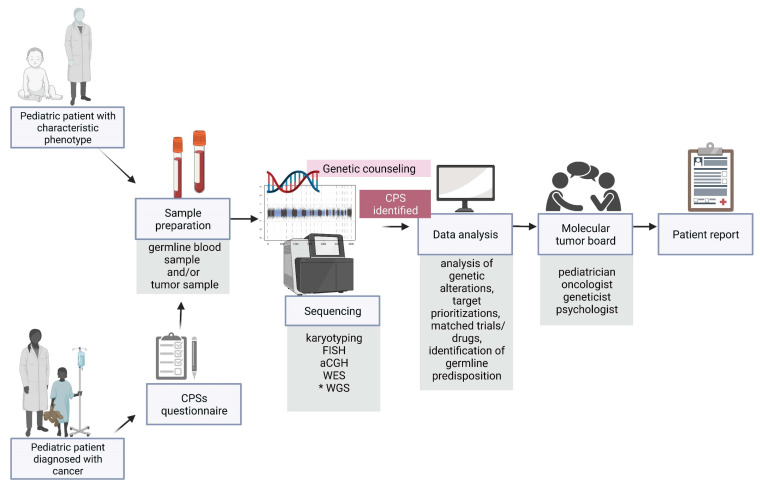
Overview of diagnostic and therapeutic approaches, highlighting a great importance of the early diagnosis of germline predisposition to paediatric neoplasms. CPS—cancer predisposition syndrome, FISH—fluorescence in situ hybridization, aCGH—array comparative genomic hybridization, WES—whole exome sequencing, WGS—whole genome sequencing. *** the genetic examination not performed routinely.** Image created with BioRender.com, accessed on 13 April 2022.

**Table 1 cancers-14-03569-t001:** Cancer predisposition syndromes with the need of more than haematological care.

Syndrome	Patient Care	References
Li Fraumeni syndrome	Children from birth to age 18: Abdominal and pelvis US every 3–4 m Annually brain MRI (first MRI with contrast) WBMRI annually	[126]
Neurofibromatosis 1	Since birth to age 8: ophthalmology assessment every 6 m to age 1 y Since age 8–20: ophthalmology assessment every 1–2 y At age of 16–20: consider WBMRI	[75]
Constitutional mismatch repair deficiency	Since age 6 WBMRI annually From diagnosis of brain tumours brain MRI every 6 m Since age 4 to 6 upper gastrointestinal endoscopy, visual capsule endoscopy, ileocolonoscopy annually	[33]
Bloom syndrome	At age 15 colonoscopy every 2 y, faecal occult blood every 6 m At age 20–25 breast MRI/US every 2 y	[28]
Xeroderma pigmentosum	Every 3 m skin examination Every 6–12 m exam for ocular and ear, nose, and throat neoplasms	[28]

m—months, y—years, WBMRI—whole-body magnetic resonance imaging, US—ultrasound, MRI—magnetic resonance imaging.

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
