# Peer review of "Genetic Disorders with Predisposition to Paediatric Haematopoietic Malignancies—A Review"

_cancers, 2022, doi:10.3390/cancers14153569_

Round 1

Reviewer 1 Report

Dear Authors,

The manuscript is you sumbitted is interesting and a relevant contribution to the field of pediatric cancer.

There is a major concern about usage of articles and prepositions in the paper, along with several grammar imprecisions, which hampers a smooth reading and for which I would suggest a revision by a native speaker.

Regarding the content of the text, I would ask you, please, to clarify the following points:

  • Page 2, line 83: "expectation" is problably " exception"?
  • Page 4, lines 182-184: " However,...": please, reformulate the sentence.
  • Page 6, line 280: "internment" is probably "intermittent"?
  • Page 9, line 475: "when affected results in tumorigenesis"? Please, clarify.
  • Page 10, line 535: "that factors." Please, clarify.
  • Page 13, line 607: "by consideration inclusion". Please, clarify.
  • Page 13, line 632: "to make properly the resulting algorithm". Please, clarify.
  • Page 14, line 657: "planned stem-cells". Please, clarify.
  • Page 14, line 658: "inflenced the selction of donor of allo-HSCT". Please, clarify.

I hope these suggestions can be helpful to improve on the manuscript.

Reviewer 2 Report

The text by Filipiuk et al. it is very interesting and reviews the genetically determined pathologies together with their predisposition to the development of tumors in pediatric age.
The authors initially describe the genetic pathologies most commonly associated with the risk of cancer development and subsequently the strategies for surveillance and early diagnosis of tumors.
Authors should also include haematological tests in the design of the follow-up examinations indicating the most appropriate timing
Authors should also present a table with acronyms present throughout the text.

Round 2

Reviewer 2 Report

The authors full replied to all criticism